



# Glaciation of Mixed-Phase Clouds: Insights from Bulk Model and Bin-Microphysics Large-Eddy Simulation Informed by Laboratory Experiment

Aaron Wang[1], Steve Krueger[2], Sisi Chen[3], Mikhail Ovchinnikov[1], Will Cantrell[4], and Raymond A. Shaw[4]

[1]Pacific Northwest National Laboratory, Richland, Washington, USA
[2]The University of Utah, Salt Lake City, Utah, USA
[3]National Center for Atmospheric Research, Boulder, Colorado, USA
[4]Michigan Technological University, Houghton, Michigan, USA

**Correspondence:** Aaron Wang (aaron.wang@pnnl.gov)

**Abstract.** Mixed-phase clouds affect precipitation and radiation forcing differently from liquid and ice clouds, posing greater challenges to their representation in numerical simulations. Recent laboratory experiments using the Pi Cloud Chamber explored cloud glaciation conditions based on increased injection of ice nucleating particles. In this study, we use two approaches to reproduce the results of the laboratory experiments: a bulk scalar mixing model and large-eddy simulation (LES) with bin microphysics. The first approach assumes a well-mixed domain to provide an efficient assessment of the mean cloud properties for a wide range of conditions. The second approach resolves the energy-carrying turbulence, the particle size distribution, and their spatial distribution to provide more details. These modeling approaches enable a separate and detailed examination of liquid and ice properties, which is challenging in the laboratory. Both approaches demonstrate that, with an increased ice number concentration, the flow and microphysical properties exhibit the same changes in trends. Additionally, both approaches show that the ice integral radius reaches the theoretical glaciation threshold when the cloud is subsaturated with respect to liquid water. The main difference between the results of the two approaches is that the bulk model allows for the complete glaciation of the cloud. However, LES reveals that, in a dynamic system, the cloud is not completely glaciated because liquid water droplets are continuously produced near the warm lower boundary and subsequently mixed into the chamber interior. These results highlight the importance of the ice mass fraction in distinguishing the mixed phase and ice clouds.

## 1 Introduction

As one of the most uncertain components in numerical weather predictions and climate models, clouds can be mixed phase (i.e., containing both supercooled liquid water and ice), making them more challenging to represent (e.g., Prenni et al., 2007; Storelvmo et al., 2008; Furtado et al., 2016; Bodas-Salcedo et al., 2016; Vergara-Temprado et al., 2018; Vignon et al., 2021; Morrison et al., 2012). Mixed phase clouds significantly affect the radiative budget (e.g., Dong and Mace, 2003; Bodas-Salcedo et al., 2016; Vergara-Temprado et al., 2018) and precipitation efficiency (e.g., Mülmenstädt et al., 2015; Field and Heymsfield, 2015); these influences differ from those of liquid water or ice clouds (Korolev et al., 2017). Additionally, aerosols can act as



cloud condensation nuclei (CCNs) or ice nucleating particles (INPs), further complicating the properties of mixed-phase clouds (e.g., Prenni et al., 2007; Tao et al., 2012; Fan et al., 2017; Fu and Xue, 2017; Norgren et al., 2018). As such, understanding the mechanisms by which a mixed-phase cloud is maintained is crucial.

An enduring challenge has been to understand how it is possible for stratiform mixed-phase clouds to persist in near steady-state microphysical conditions despite the fact that the supercooled liquid is in a metastable state (e.g., Fridlind et al., 2007; Westbrook and Illingworth, 2013; Yang et al., 2013; Ovchinnikov et al., 2014, 2011; Solomon et al., 2018; Fu et al., 2019). Various factors contribute, including steady forcing by radiative cooling or surface heating, and maintaining a source of ice through the steady generation of ice via stochastic nucleation, the entrainment of ice nucleating particles from above, etc. A

fundamental aspect is the strength of forcing of supersaturation in a cloud and the amount of ice needed to reduce the liquid-water supersaturation to less than zero such that the Wegener–Bergeron–Findeisen process can occur (Korolev, 2007; Korolev et al., 2017). Thus, the relative concentrations of cloud droplets and ice particles play a central role in the glaciation or lack thereof within a mixed-phase cloud.

Glaciation refers to the transition from a mixed-phase cloud to an ice-only cloud. This transition occurs because, under a

given water vapor pressure, the supersaturation with respect to ice is greater than that with respect to liquid water. When ice and liquid particles coexist, ice gains mass from the droplets indirectly via deposition of vapor evaporated from the droplets. This mechanism is known as the Wegener–Bergeron–Findeisen process (Wegener, 1911; Bergeron, 1928; Findeisen, 1938). Nevertheless, there is no consensus on the quantitative condition to distinguish the mixed-phase and ice clouds (Korolev et al., 2017). From a theoretical perspective, Korolev and Mazin (2003) defined glaciation as the reduction of the mean supersaturation

with respect to liquid to below zero. However, in reality, fluctuations in supersaturation may still produce droplets even when the mean value is below zero (Prabhakaran et al., 2020). In field measurements, the ice phase mass fraction, i.e., the ratio of ice mass to the sum of liquid and ice mass, which varies between 0 and 1, is often used to define the cloud phase (Korolev et al., 1998), but then the thresholds to differentiate between liquid, mixed-phase, and ice clouds become an important consideration. Many studies use 0.9 as a threshold for the ice mass fraction to distinguish between mixed-phase and ice-phase clouds (e.g.,

Korolev et al., 2003; Field et al., 2004), yet there is no physical basis for the value used (Korolev et al., 2017).

The investigation of mixed-phase clouds has relied on field campaigns and numerical simulations (e.g., Curry et al., 1997; Pinto, 1998; Korolev et al., 2003; Field et al., 2004; Verlinde et al., 2007; Fridlind et al., 2012; Hill et al., 2014; Ovchinnikov et al., 2014; Pinsky et al., 2018; de Roode et al., 2019; Morrison et al., 2011). However, direct comparisons of field measurements with numerical simulations are difficult owing to the poorly constrained initial and boundary conditions for models as

well as the difficulties of measuring the properties of mixed-phase clouds. The recent development of the Pi Cloud Chamber has provided well-defined boundary conditions to produce steady convection clouds and furnished detailed microphysical measurements (Chang et al., 2016; Chandrakar et al., 2016), allowing for a better comparison with numerical simulations (Thomas et al., 2019; Chen et al., 2024). Experiments on mixed-phase clouds have also been conducted by Desai et al. (2019) using the Pi Cloud Chamber. Specifically, Desai et al. (2019) tested how many INPs are needed to glaciate a mixed-phase cloud. They

found that the injection rate of INPs determines the ice water content and thus the ratio of ice to total water content. Their





measurements were shown to be consistent with the theoretical value of ice integral radius needed for glaciating a mixed-phase cloud proposed by Korolev and Isaac (2003).

Guided by the laboratory experiments of Desai et al. (2019), we employ two approaches to explore aerosol-mediated mixed-phase cloud. First, we use a bulk scalar mixing model (which is referred to as the bulk model in this study) to solve for the domain-average properties. The bulk model is similar to the scalar flux budget model (Thomas et al., 2019; Yeom et al., 2023; Wang et al., 2024a; Chen et al., 2024) or mean-field model (Shaw et al., 2023) applied in the previous studies, but includes both liquid and ice condensate components. The efficiency of the bulk model is crucial, as it provides quick results over a continuous range of input parameters. Second, we perform large-eddy simulations (LESs) with bin microphysics for aerosols, liquid water droplets, and ice crystals. Different from the bulk model, LES resolves the dynamics of energy-carrying turbulence, and bin microphysics resolves the particle size distributions (PSDs), so the combination provides more details of the mixing process and PSD development. We will compare the model-produced statistics with the measurements made by Desai et al. (2019), alongside the theoretical value of ice integral radius proposed by Korolev and Isaac (2003), and analyze quantities that are challenging to measure in the laboratory. Lastly, we will explore how the ice mass fraction evolves in both modeling approaches when glaciation occurs.

The rest of the paper is organized as follows: Section 2 details the methods. Section 3 presents the results. Section 4 concludes the paper. Additionally, since the statistics of microphysical properties are based on the cutoff radius used in the laboratory experiments, Appendix A explores the sensitivity to the cutoff radius. Lastly, Appendix B provides some examples to illustrate how the bulk model can be applied to quickly evaluate an experiment setup.

## 2 Methods

### 2.1 The Bulk Scalar Mixing Model

The bulk model is composed of the budget equations for two scalars (temperature and water vapor mixing ratio) and assumes that they are well-mixed (uniform) within the domain, which is a good assumption for Rayleigh-Bénard convection except very near to the walls. In a rectangular cuboid chamber domain, the budget equation for temperature is

$$\frac{dT}{dt} = \frac{1}{h}\left(F_{T,b} + F_{T,t} + \frac{A_s}{A_b}F_{T,s}\right) + \frac{L}{c_p}\tilde{c} + \frac{L_d}{c_p}\tilde{d}, \tag{1}$$

where $T$ denotes temperature, $F_{T,b}$, $F_{T,t}$, and $F_{T,s}$ are the temperature fluxes per unit area into the chamber at the bottom, top, and side walls, $h$ is the chamber height, $A_b$ is the area of the bottom (also top) wall, $A_s$ is the total area of the side walls, $\tilde{c}$ is the net condensation rate for the growth of droplets (kg kg$^{-1}$ s$^{-1}$), $\tilde{d}$ is the net deposition rate for the growth of ice crystals (kg kg$^{-1}$ s$^{-1}$), $L$ is the latent heat of condensation, $L_d$ is the latent heat of deposition, and $c_p$ is the heat capacity of air. The chamber height and wall areas in our calculations are based on a domain of 2 m by 2 m by 1 m, which is the same as that used in Thomas et al. (2019) and the LES setup in this study.





The fluxes are calculated using a bulk aerodynamic formulation:

$$F_{T,b} = V_b(T_b - T), \tag{2}$$

$$F_{T,t} = V_t(T_t - T), \tag{3}$$

$$F_{T,s} = V_s(T_s - T), \tag{4}$$

where $T_b$, $T_t$, and $T_s$ are the bottom-, top-, and side-wall temperatures, and $V_b$, $V_t$, and $V_s$ are the effective eddy velocities for bottom, top, and side walls, respectively. Following Desai et al. (2019), we set $T_b$, $T_t$, and $T_s$ to $4°C$, $-16°C$, and $-12°C$, respectively. Regarding the effective velocity, because the direction of buoyancy is normal to the bottom and top walls, a previous LES study of cloud chambers suggests that $V_s$ is less than $V_b$ and $V_t$ (Wang et al., 2024a). Here we assume that $V_b = V_t = 4$ mm s$^{-1}$ (estimated based on the Rayleigh number and Nusselt number according to Niemela et al., 2000), and the

velocity ratio of $V_s$ to $V_b$ is assigned a value of 0.42 based on the LES wall fluxes in this current study. Note that this ratio was found to be 0.35 in Wang et al. (2024a), and the larger value used here implies a slightly greater relative influence of side walls compared to the bottom and top walls.

The budget equation for water vapor mixing ratio, $q_v$, is similar to that for $T$:

$$\frac{dq_v}{dt} = \frac{1}{h}\left(F_{q_v,b} + F_{q_v,t} + \frac{A_s}{A_b}F_{q_v,s}\right) - \tilde{c} - \tilde{d}, \tag{5}$$

where $F_{q_v,b}$, $F_{q_v,t}$, and $F_{q_v,s}$ are the corresponding water vapor fluxes per unit area into the chamber. They are calculated in the same way as those for temperature in Eqs. 2–4. The values of $q_v$ at the walls are saturated with respect to liquid (bottom) or ice (side and top walls). For the side walls, in addition to being saturated, we will also test a side-wall wetness of 0.30 (i.e., the boundary condition of $q_v$ corresponds to a saturation ratio of 0.30 with respect to ice at the side-wall temperature) which is used in the LES setup described in Section 2.2.

The condensation and deposition rates for uniform supersaturation were derived by Korolev and Mazin (2003)

$$\tilde{c} = \frac{\rho_l}{\rho} 4\pi \, s_l \xi_l N_l \bar{r}_l \tag{6}$$

and

$$\tilde{d} = \frac{\rho_i}{\rho} 4\pi \, s_i \xi_i N_i \bar{r}_i \tag{7}$$

where $\rho$ is the air density, $\rho_p$ is the particle density, $s_p$ is the supersaturation with respect to the phase of the particle, and $\bar{r}_p$

is the particle mean radius. The subscript $p$ indicates whether the particle is liquid ($l$) or ice ($i$). $\xi_l$ and $\xi_i$ are the condensation and deposition parameters, respectively, which are functions of pressure and temperature (Rogers and Yau, 1996).

Lastly, for uniform supersaturation and no curvature or solute effects on particle growth, the mean equilibrium radii for droplets and ice crystals are given by Krueger (2020):

$$\bar{r}_l = \frac{\sqrt{2}\Gamma(\frac{3}{4})}{\sqrt{\pi}}\hat{r}_l, \tag{8}$$

$$\bar{r}_i = \frac{\sqrt{2}\Gamma(\frac{3}{4})}{\sqrt{\pi}}\hat{r}_i, \tag{9}$$



which are defined in terms of the corresponding mode radii:

$$\hat{r}_l \equiv \left(\frac{s_l \xi_l h}{k_1}\right)^{\frac{1}{4}}, \tag{10}$$

$$\hat{r}_i \equiv \left(\frac{s_i \xi_i h}{k_1}\right)^{\frac{1}{4}}, \tag{11}$$

where $k_1 = (\rho_p/\rho_l)1.19 \times 10^8 \text{ m}^{-1} \text{ s}^{-1}$ is the coefficient of Stokes drag (Rogers and Yau, 1996). We assume that ice particles
are spherical and that $\rho_i = 900 \text{ kg m}^{-3}$.

We solve for the equilibrium values of temperature and water vapor using (1) and (5) given specified number concentrations of droplets and ice, $N_l$ and $N_i$. Specifically, we set $N_l$ as 25 cm$^{-3}$, as measured in Desai et al. (2019) before the existence of ice, and vary $N_i$ to mimic the effect of injecting ice nucleating particles at various rates in the laboratory chamber. The temperature weakly affects $\xi_1$. The chamber height, particle density, radius, and supersaturation combine to determine the
equilibrium mean radii of the droplets and ice particles according to (8) and (9). Combined with supersaturations over liquid and ice, we can also calculate the condensation and deposition rates using (6) and (7), which then influence the temperature and water vapor mixing ratio values. Their equilibrium values are obtained by iteration.

## 2.2   Large-Eddy Simulation with Bin Microphysics

Following the previous LES studies on the Pi Cloud Chamber (Thomas et al., 2019, 2023; Yang et al., 2022, 2023; Wang et al.,
2024a, in review), we use the System for Atmospheric Modeling (SAM, Khairoutdinov and Randall, 2003) for the simulations, but in this case we extend it to include ice microphysics. In SAM, the velocity equations are solved on an Arakawa staggered C-grid (Arakawa and Lamb, 1977). The velocity is advected using a second-order central scheme and dissipated by the turbulent kinetic energy (TKE) subgrid-scale model (Deardorff, 1980). The scalars are solved at the centers of the grid cells, advected by a multidimensional positive definite advection transport algorithm (Smolarkiewicz and Grabowski, 1990), and diffused with a
turbulent Prandtl number of 1 (i.e., Reynolds analogy, Kays et al., 1980).

The domain is 2 m by 2 m by 1 m with a uniform grid spacing of 3.125 cm in all directions, which falls within the inertial subrange according to the direct numerical simulations performed at similar Reynolds numbers (Wang et al., 2024b). The velocity boundary conditions are for a solid (no-slip and no-penetration). The surface shear stress, sensible heat flux, and moisture flux are solved based on Monin-Obukhov similarity theory. The roughness lengths for velocity, temperature, and
moisture are set to 0.75 mm, 0.46 mm, and 0.57 mm, respectively, as explored and detailed by Wang et al. (2024a). Following the experimental setup in Desai et al. (2019), the temperatures of the bottom wall, top wall, and side walls are set to 4°C, -16°C, and -12°C, respectively. Regarding the wetness, the bottom wall is saturated with respect to liquid water, and the top wall is saturated with respect to ice. The side walls are complex, as they are partly ice-covered and partly clear (e.g., windows) in the actual Pi Cloud Chamber. Previous LES studies of the Pi Cloud Chamber (Thomas et al., 2019; Wang et al., 2024a) tuned
the side-wall wetness to match the supersaturation inferred in the laboratory. In this study, we tune the side-wall wetness with respect to ice and aerosol injection rate to match the droplet size and number concentration measured by Desai et al. (2019).





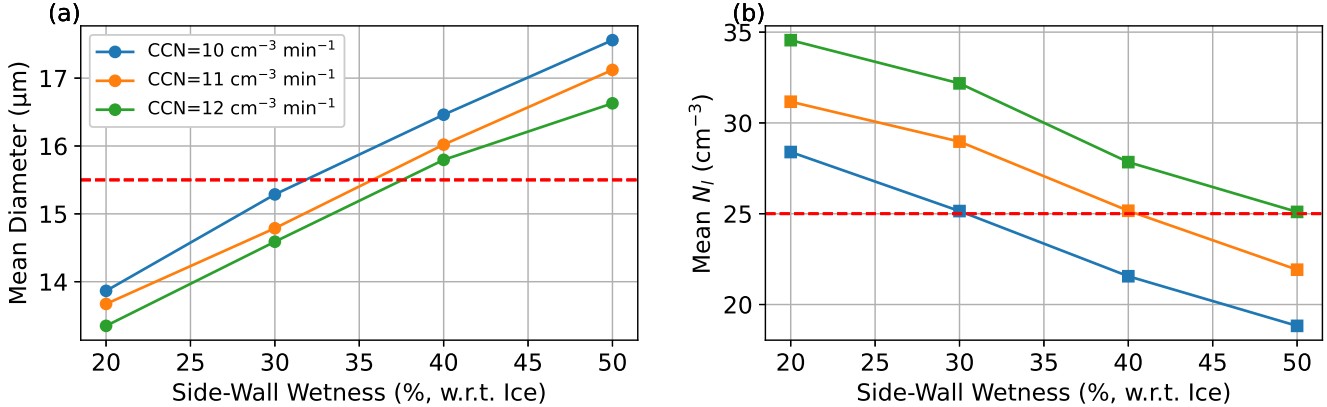

**Figure 1.** LES results without ice for tuning the CCN injection rate and side-wall wetness to match the diameter (a) and number concentration (b) from the experiments. Each data point represents a steady-state result of one simulation. The red dashed lines represent the observations in the Pi Cloud Chamber that we aim to match.

In order to match the steady-state diameter of 15.5 $\mu$m and number concentration of 25 cm$^{-3}$ as observed in the Pi Cloud Chamber by Desai et al. (2019), we perform simulations with various combinations of CCN injection rates and side-wall wetness. The CCN, NaCl, is homogeneously and continuously injected within the domain. Each simulation runs for 20 min, and

averages of the last 5 min are used for analysis. Note that these simulations do not contain ice; they are only used to determine the setup for the later mixed-phase simulation that explores the glaciation mechanism. Figure 1 demonstrates that SAM LES can achieve a diameter of 15.3 $\mu$m and a number concentration of 25.1 cm$^{-3}$ when CCN injection rate is 10 cm$^{-3}$ min$^{-1}$ and side-wall wetness is 0.30. These CCN injection rate and side-wall wetness values are thus used for the mixed-phase simulation. For comparison, in the Pi Cloud Chamber experiment by Desai et al. (2019), if the CCN were evenly distributed throughout to

the entire cylindrical chamber, their CCN injection rate would be 12.7 cm$^{-3}$ min$^{-1}$.

The bin microphysics model applied here is developed by the Hebrew University of Jerusalem group (Khain et al., 2004) and has also been employed in studying the Pi Cloud Chamber experiment by Thomas et al. (2019, 2023); Yang et al. (2022); Wang et al. (2024a). Both droplets and ice crystals are represented by 33 mass-doubling bins. Their fall speeds mainly follow the Stokes law for diameters smaller than 40 $\mu$m (Rogers and Yau, 1996, Section 8.a). Collision-coalescence and the collision

involving ice are disabled, as the time-scale analysis indicates that the sedimentation time is much shorter than the collision time in such domain (Chandrakar et al., 2016; Shaw et al., 2020). The ice crystals are assumed to be spherical owing to their small lifetime in the Pi Cloud Chamber (Desai et al., 2019). Although the smallest droplet and ice crystal in the bin microphysics have diameters of approximately 2 $\mu$m, for accurate comparison with the results measured in the Pi Cloud Chamber by Desai et al. (2019), we set the cutoff radius at 3.5 $\mu$m to calculate the microphysical properties for analysis.

To avoid the uncertainties associated with ice nucleation, and considering that Desai et al. (2019) used highly efficient ice nucleating particles (Snomax, Lukas et al., 2022), we inject the smallest ice that the bin microphysics can resolve (roughly 2 $\mu$m in diameter) directly in the LES. The ice is injected into the central top four grid cells to mimic the spreading of



Snomax from a port in the top of the Pi Cloud Chamber. We perform the whole simulation for 90 min, during which the initial
20 min are without ice injection and only with aerosol injection, allowing the supercooled droplets to reach a steady state.
Subsequently, the ice injection rate is gradually increased every 10 min in the following sequence: 0.5, 1.5, 3.0, 5.0, 7.5, 10.0,
and 15.0 cm$^{-3}$ min$^{-1}$. These values represent the rate of ice injection per minute averaged over the entire domain. The flow
reaches a new equilibrium roughly within 5 min, so we take the temporal average of the final 5 min of each 10-min interval to
assess the droplet and ice PSDs. Note again that this mixed-phase simulation, performed for 90 min, differs from the multiple
20-min liquid-phase simulations previously used to determine the CCN injection rate and side-wall wetness.

## 2.3   Theoretical Threshold of Ice Integral Radius to Glaciate a Mixed-Phase Cloud

Desai et al. (2019) proposed a theoretical condition for the ice integral radius in order for a mixed-phase cloud to exist. This
condition on the integral radius is analogous to that derived by Korolev and Mazin (2003) in terms of critical updraft velocity.
Specifically, Eq. 4 in Desai et al. (2019) (which is analogous to Eq. 4 in Korolev, 2007) demonstrates the condition of ice
integral radius for a mixed-phase cloud as:

$$n_i \bar{r}_i \leq \frac{1}{4\pi D \tau_t} \frac{s_{1,0}}{s_1^\star}, \tag{12}$$

where $D$ is the molecular diffusion coefficient; $\tau_t$ is the turbulent mixing time for the flow properties to relax to an equilibrium
state; $s_{1,0}$ is the initial supersaturation of liquid without aerosols; $s_1^\star$ is the liquid-water supersaturation deficit that exists at the
ice-saturation level, which is calculated as

$$s_1^\star = -\frac{e_i - e_l}{e_l} = \frac{s_i - s_l}{s_i + 1}, \tag{13}$$

where $e_i$ and $e_l$ are the saturation vapor pressures for ice and liquid water, respectively.

Most of the values in Eq. 12 can be obtained from the bulk model or LES output. $D$ is interpolated from Table 7.1 in Rogers
and Yau (1996) using the mean temperature. To obtain $s_{1,0}$, the bulk model simply solves Eqs. 1 and 5 without condensation
and deposition terms, resulting in a steady-state $s_{1,0}$ of 5.6% (when side-wall wetness is 0.30) or 19% (when side walls are
saturated). For LES, an additional moist simulation with side-wall wetness of 0.30 is conducted without ice and CCN, resulting
in a steady-state $s_{1,0}$ of 5.97%. Regarding $\tau_t$ for the bulk model, Eqs. 1–5 are rearranged as follows:

$$\frac{d\phi}{dt} = \frac{1}{\tau_t} \left( \frac{\phi_b + \phi_t + \widetilde{A}\widetilde{V}\phi_s}{2 + \widetilde{A}\widetilde{V}} - \phi \right), \tag{14}$$

where $\widetilde{A} = A_s/A_b = 2$ for the studied domain, $\widetilde{V} = V_s/V_b$ is 0.42 in this study, and $\tau_t = h/[(2 + \widetilde{A}\widetilde{V})V_b] = 88.0$ s. For LES,
the above-mentioned moist simulation, which has already reached a steady-state $q_v$ ($q_{v,1}$), is performed for an additional 10 min
with the wetness of all walls reduced by 20% to approach a new steady state ($q_{v,2}$). The time taken to reach $q_{v,2} + (q_{v,1} - q_{v,2})/e$
is used as $\tau_t$ for LES, which is 85.7 s. For comparison, the values used in Desai et al. (2019) are $s_{1,0} = 3.1\%$ and $\tau_t = 61.2$ s.



## 3   Results

### 3.1   The Bulk Scalar Mixing Model

The results (mean temperature, water vapor mixing ratio, condensation or deposition rate, supersaturation, mean particle radius, number concentration, water content, and ice integral radius) predicted by the bulk model are plotted in Fig. 2. The ice-phase-related properties are represented by blue colors, the liquid-phase-related properties are displayed in red colors, and the line styles differentiate the side-wall wetness.

Glaciation occurs in the bulk model when the supersaturation over liquid decreases to zero, which occurs when $N_i$ increases to 3 to 10 cm$^{-3}$ (depending on the sidewall wetness, SW). Before complete glaciation, the deposition rate increases (Eq. 7) while the condensation rate decreases. The net effects of the latent heat release due to deposition and condensation result in a slight increase in temperature (Eqs. 1), and a slight decrease in the supersaturation with respect to both ice and liquid water (Fig. 2d). The decrease in supersaturation also reduces the radii of droplets and ice crystals (Eqs. 8–9; Fig. 2e) and eventually lowers the condensation rate to zero (Eq. 6; Fig. 2c) as the supersaturation over liquid becomes negative. Glaciation occurs when droplets can no longer form, which is indicated by zero values in condensation rate, supersaturation over liquid, droplet radius, liquid water content (Fig. 2c, d, e, f), although the droplet number concentration is assumed to be constant in the bulk model (Fig. 2g). Without the replenishment from droplet evaporation, the water vapor tends towards saturation over ice more rapidly, enhancing the decrease in ice crystal size (Fig. 2b, d, e). As the number of ice crystals increases to compete for water vapor, each ice crystal obtains less water vapor for growth, leading to an increase in the total mass but a decrease in the mean size. The trends in supersaturation and water contents agree with Fig. 7 in Korolev and Mazin (2003). Additionally, the ice integral radius for glaciation predicted by Korolev and Mazin (2003) matches those predicted by the bulk model (Fig. 2h), as both approaches assume a well-mixed cloud. When the side-wall wetness is reduced, glaciation occurs earlier because of the reduced supersaturation with respect to liquid water due to lower vapor fluxes from the walls.

For comparison, the LES results, which will be discussed in detail later, are mapped onto the bulk-model results. LES agrees well with the bulk model in temperature, water vapor mixing ratio, supersaturation, and ice properties (Fig. 2a, b, d–f, h). However, the bulk model predicts that the mixed-phase cloud can be completely depleted because the domain is always well-mixed. In contrast, LES predicts that the mixed-phase cloud is not completely glaciated because the domain is not well-mixed, so some regions can remain liquid-saturated even though the mean liquid-supersaturation is negative.

### 3.2   Large-Eddy Simulation with Bin Microphysics

To provide a comprehensive view of the flow within the domain, Fig. 3 presents a snapshot at $t = 50$ min, illustrating how the LES captures the detailed processes of turbulence mixing in this dynamic system. This demonstrates that the chamber is in fact not perfectly well mixed, in spite of the success of the bulk model. The large-scale circulation in the chamber forms a single roll, characterized by an updraft of warm air (indicated by the red-to-yellow isosurface in Fig. 3) and a downdraft of cold air on the opposite side of the domain (not shown). The updraft region exhibits a higher supersaturation during the mixing process, resulting in an increase in the liquid water and ice water contents above the warm updraft (indicated by the purple-to-white





**Figure 2.** Results of the bulk model for (a) temperature, (b) water vapor mixing ratio, (c) condensation and deposition, (d) supersaturation, (e) mean radius, (f) water content (WC), (g) number concentration, and (h) ice integral radius under various side-wall wetness (SW, with solid line representing saturated side walls and dashed line representing a SW of 0.30). Blue colors indicate ice properties, whereas red colors represent liquid properties. The purple lines in panel (h) denote the glaciation threshold suggested by Korolev and Mazin (2003) for various side-wall wetness. The LES results are mapped for comparison (markers with uncertainty bars representing 5 to 95 percentiles of the spatial distribution) except for panel (c), as three-dimensional condensation and deposition data are not output in the LES. The corresponding ice injection rates for the mapped LES data are indicated on the top of the panels as shown in Fig. 4.





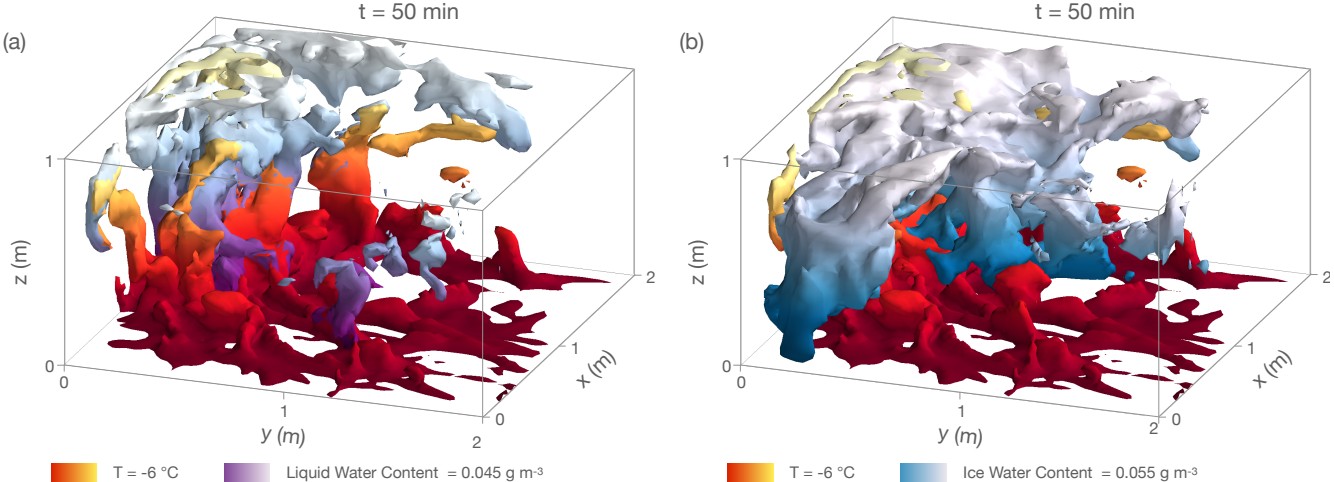

**Figure 3.** A snapshot of warm air isosurface of $T = -6°C$ [the red-to-yellow color in panels (a) and (b)], liquid water content isosurface [the purple-to-white color in panel (a)], and ice water content isosurface [the blue-to-white color in panel (b)] at $t = 50$ min. Gradient colors represent different heights, with lighter colors indicating closer to the chamber top, to enhance the three-dimensional visualization. The isosurfaces are opaque, causing some warm air isosurfaces to be obscured by the liquid water and ice crystal isosurfaces.

isosurface in Fig. 3a and the blue-to-white isosurface in Fig. 3b, respectively). As the warm updraft mixes with the cold air near the chamber's top wall, the liquid water and ice water contents gradually decrease. On the downdraft side, the supersaturation is lower compared to the updraft side, because the temperature difference between the side walls and top wall is less than that between the side walls and the bottom wall. Note that in the absence of any horizontally asymmetric forcing, the direction of the large-scale circulation varies over time, as also demonstrated by observational studies in the Pi Cloud Chamber (Anderson et al., 2021) and numerical simulations (Huang and Zhang, 2023; Wang et al., 2024a).

Figure 4 presents the time series of the flow properties (temperature, water vapor mixing ratio, supersaturation, particle number concentration, particle mean radius, water content, ice mass fraction, and ice integral radius) averaged over the domain, excluding the regions within 6.25 cm from each wall. The shading areas show the 5 to 95 percentiles of the spatial distributions to indicate the variability of the properties within the domain. Figure 4a shows that the temperature increases slightly with the ice injection rate because of the enhanced latent heat release from ice deposition. This increase is more pronounced after $t = 40$ min, corresponding to an ice injection rate of $3.0\,\mathrm{cm^{-3}\,min^{-1}}$, which is illustrated more clearly in Fig. 2a. Additionally, the simulated mean temperature before injecting ice is close to the observed value of $-6.7°C$ reported in Desai et al. (2019). The water vapor mixing ratio shows a slight decrease after $t = 50$ min, corresponding to an ice injection rate of $5.0\,\mathrm{cm^{-3}\,min^{-1}}$ (Figure 4b), and this trend agrees with the bulk model (Fig. 2b). Figure 4c demonstrates that the supersaturation with respect to liquid and ice decreases with the increasing injection of ice. The rate of decrease accelerates after $t = 40$ min, when the mean supersaturation with respect to liquid falls below zero. After $t = 50$ min, most of the domain starts to become subsaturated with respect to liquid (i.e., most of the interquartile range also falls below zero). All of these observations are related to the







**Figure 4.** The time series of (a) temperature, (b) water vapor mixing ratio, (c) supersaturation, (d) particle number concentration, (e) particle mean radius, (f) water content (WC), (g) ice mass fraction, and (h) ice integral radius. The shading areas show the 5 to 95 percentiles of the spatial distribution to represent the uncertainties. The uncertainties for the total values (gray lines) are not displayed in panels (d) and (f) to optimize the clarity of the plot. The black dashed line in panel (g) indicates the previously applied glaciation threshold (i.e., ice mass fraction $> 0.9$). The purple dashed line in panel (h) represents the theoretical condition, according to Eq. 12, below which the mixed phase clouds are hypothesized to exist. The time series displays periods of different ice injection rates, with the injection rate increasing from 0 to $15\ \mathrm{cm}^{-3}\ \mathrm{min}^{-1}$, as shown on the top of the panels.

occurrence of glaciation during $t = 40\text{–}60$ min. The ice injection rate during $t = 50\text{–}60$ min is $5.0\ \mathrm{cm}^{-3}\ \mathrm{min}^{-1}$, consistent with the Snomax injection rate that causes glaciation in Desai et al. (2019).

When ice is first injected into the domain at $t = 20$ min, the droplet number concentration slightly increases at first because there is a slowdown in droplet sedimentation due to the reduced droplet size (Fig. 4d, e). After $t = 40$ min, the droplet number concentration starts to drop rapidly because the mean supersaturation with respect to liquid falls to below 0 (Fig. 4c). During $t = 50\text{–}60$ min, when most of the domain is subsaturated with respect to liquid (Fig. 4c), the number concentration of droplets



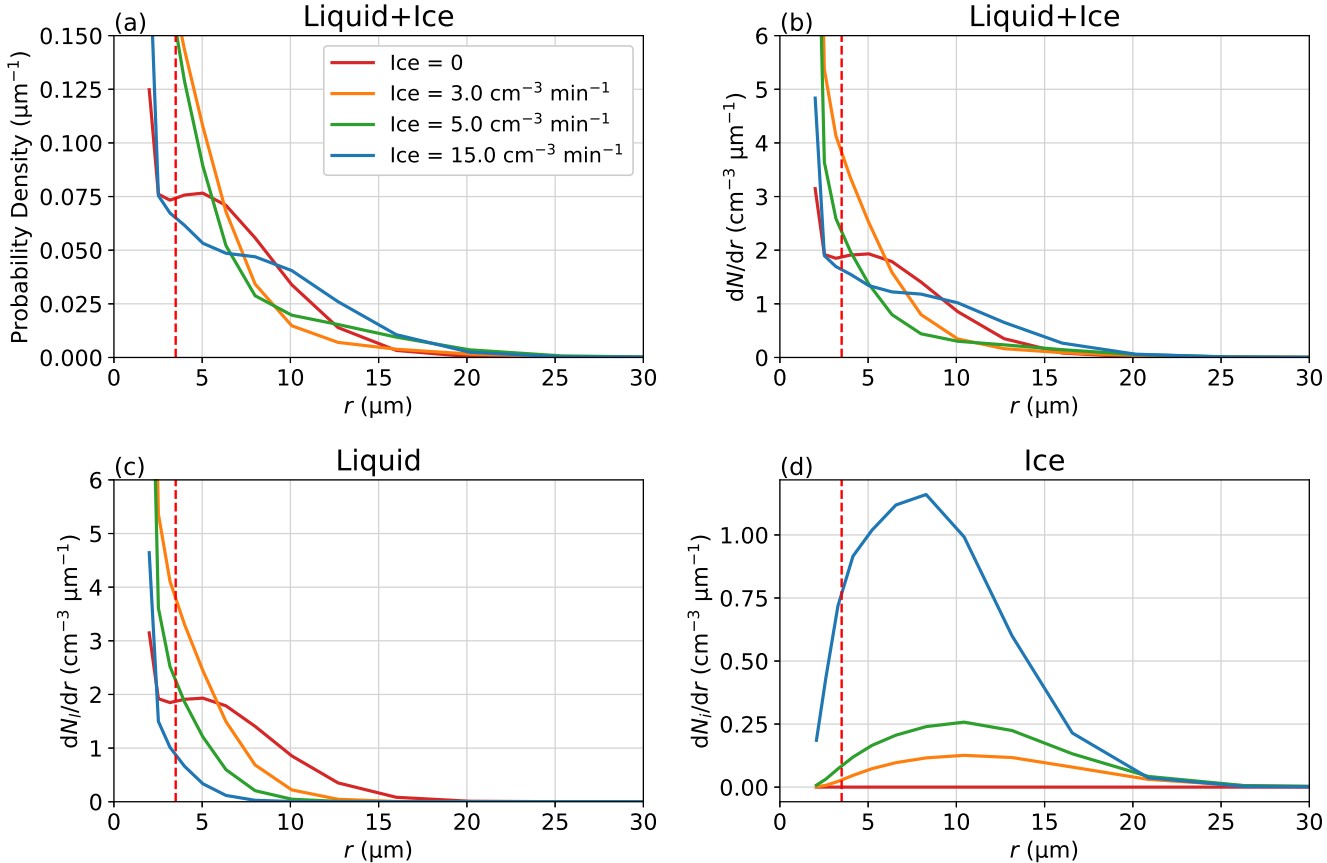

**Figure 5.** (a) The probability density function of the PSDs, including both ice and droplets (normalized by the particles larger than the cutoff radius of 3.5 $\mu$m), (b) the size distribution of drop and ice, (c) the droplet size distribution, and (d) the ice size distribution.

still exceeds that of ice crystals. However, ice crystals are consistently larger than the droplets (Fig. 4e), and the total mass of ice already exceeds that of droplets after $t = 40$ min (Fig. 4f). In fact, the ice mass fraction reaches 0.9 during $t = 50$–60 min
(marked by the black dashed line in Fig. 4g), which is the glaciation threshold suggested by Korolev et al. (2003) and used in the analysis of Desai et al. (2019).

Although the mean ice mass fraction exceeds 0.9 after $t = 50$ min, the ice integral radius has exceeded the theoretical threshold suggested by Korolev and Mazin (2003) after $t = 40$ min (Fig. 4h), when the mean supersaturation with respect to liquid is already below zero (Fig. 4c), and the rate of increase in temperature has already accelerated (Fig. 2a). During $t = 45$–
50 min, the ice mass fraction ranges from 0.36 to 0.80 with a mean value of 0.61, lower than the suggested threshold of 0.9. Regarding this, one may reconsider the threshold of ice mass fraction to define glaciation.

Unlike the laboratory-measured PSDs where distinguishing small-sized droplets and ice is challenging, LES allows us to examine the PSDs of droplets and ice separately. When ice is injected, the peak in total PSD (red curve in Fig. 5a–b) that



corresponds to the mode peak in the droplet size distribution (Fig. 5c) disapears, and another peak emerges (blue curve in

Fig. 5a–b) which corresponds to the ice mode (5d) . This ice mode peak shifts to the left as the ice injection rate increases. Owing to the numerical diffusion of the bin microphysics scheme that artifically broadens the PSDs, the total PSDs do not display two distinct peaks simultaneously as observed by Desai et al. (2019). However, the separate PSDs in LES reveal the decrease of droplet size with the increase in ice (cf. Fig. 5c, d), qualitatively agreeing with the observation by Desai et al. (2019).

Finally, in Fig. 6, we examine the cross section of the cloud and vapor fields at the end of each ice injection rate period. Overall, the supersaturation with respect to liquid (represented by the shading) decreases as ice increases (represented by the isoline), consistent with the domain-mean trend in Fig. 4c. The high ice water content (represented by the solid isoline) arises from the near-bottom region with high supersaturation (representing the updraft regions) and extends to the core region with low supersaturation. This pattern possibly implies how the water vapor in the core region is consumed by the ice. However,

regardless of the intensity of the ice injection rate, the near-bottom region always exhibits positive supersaturation with respect to liquid, resulting from the strong water vapor flux from the bottom. Due to this heterogeneous nature in supersaturation, although the domain-average properties may present the characteristics of ice clouds, liquid water is never completely depleted (as also shown in Fig. 4–5), and the determination of glaciation requires a threshold. The surviving liquid droplets are not represented by the bulk model, which assumes an always well-mixed cloud, a concept that is unlikely to be exactly true in

either the cloud chamber or atmospheric clouds.

## 4   Conclusion and Discussion

In this study, we applied two approaches to reproduce the laboratory study of the glaciation of mixed-phase clouds, which was conducted in the Pi Cloud Chamber by Desai et al. (2019). The first approach is a bulk scalar mixing model, which determines the equilibrium values of two well-mixed flow quantities within the chamber (temperature and water vapor mixing ratio), given

the top, bottom and side wall temperatures, side wall wetness, and cloud ice and cloud droplet number concentrations. The second approach is LES with bin microphysics, which resolves the main turbulent mixing within the chamber to provide more detailed insights into the glaciation process. In the LES approach, the same chamber boundary conditions used for the bulk model are imposed, but the particle injection rates are specified instead of the equilibrium particle number concentrations. These two numerical approaches enable us to study the liquid and ice properties separately (and spatially in the LES approach), which

is challenging to do in the laboratory, while also being capable of exploring a broad parameter space economically (especially with the bulk model).

    The calculations over a wide, continuous range of ice concentration provided by the bulk model reveal distinctive trends in flow quantities with the increase in ice concentration before and after glaciation occurs. Specifically, after a mixed-phase cloud is glaciated, the temperature increase in response to the increased ice concentration becomes more prominent, and the water

vapor mixing ratio decreases. This leads to a substantial decrease in the supersaturation as the ice concentration increases,





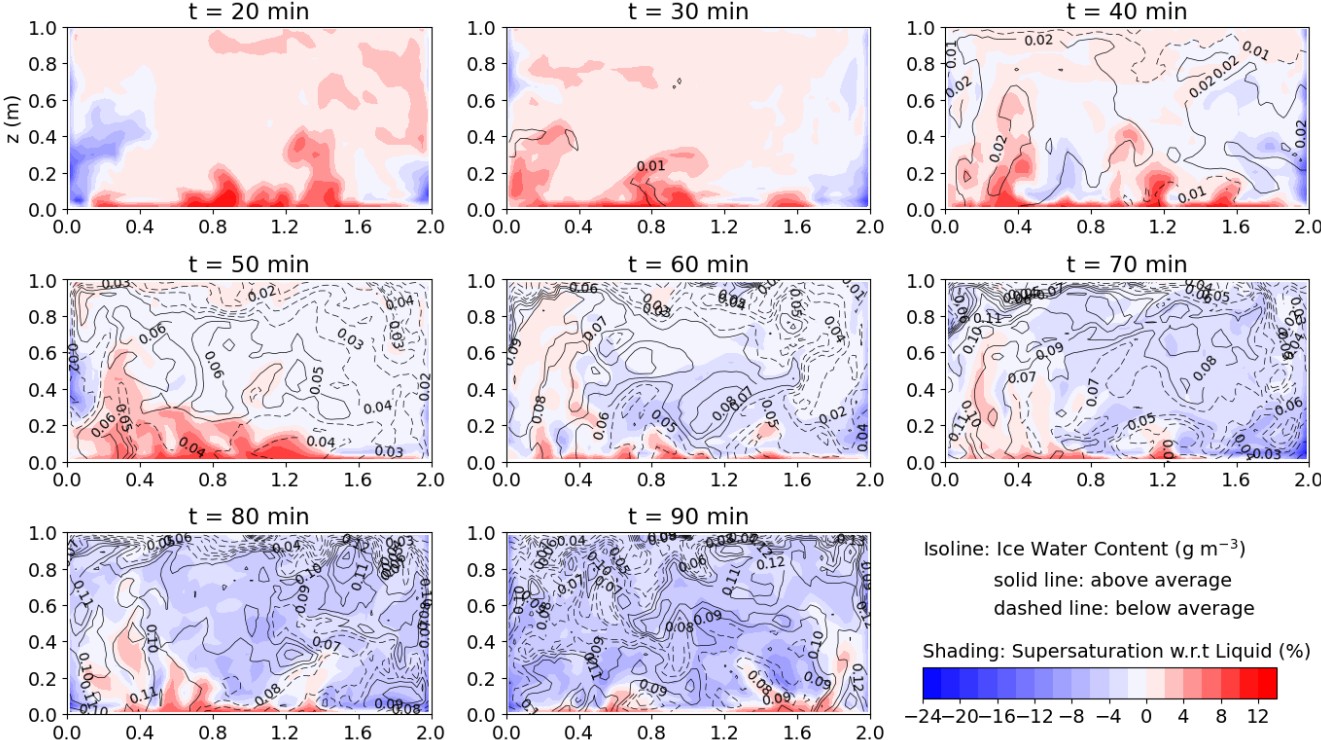

**Figure 6.** The ice mass concentration (represented by isolines; unit: $g\,m^{-3}$) where solid lines indicate above-average values, and dashed lines indicate below-average values, alongside supersaturation with respect to liquid (indicated by shading), observed at the end of each time period. Red shading represents supersaturation, while blue shading indicates subsaturation.

causing the size of the ice crystals to diminish. Such changes in trends are also observed in the LES results, although, unlike in the bulk model, the liquid water is never completely depleted in the LES.

The LES results reveal that, when the mean supersaturation with respect to liquid is below zero, the trends in temperature and supersaturation behave similarly to the bulk model, and the mean ice integral radius reaches the theoretical threshold, which

indicates that glaciation can occur. However, the ice mass fraction has a mean of 0.61 at this time, which is significantly less than the suggested threshold of 0.9. When the supersaturation with respect to liquid within most of the domain is below zero, the ice mass fraction increases to approximately 0.90. Even with the ice injection rate increased by a factor of three (from 5 to $15\,cm^{-3}\,min^{-1}$), the liquid water content is not fully depleted. This is due to the strong water vapor flux near the bottom wall that sustains the locally supersaturated environment and maintain droplets there. Thus, having a correct threshold of the ice

mass fraction for glaciation that reflects this locally heterogeneous nature of supersaturation fields turns out to be important.

This study illustrates how the bulk model and LES can be useful tools to reproduce, interpret, and extend the experiments conducted in the Pi Cloud Chamber. The results of both approaches generally agree with the experiment by Desai et al. (2019). The agreement with a rigorous, well-characterized laboratory experiment strengthens confidence that these modeling





approaches can successfully represent the physics in naturally occurring mixed phase clouds, such as long-lived Arctic stratus.
In particular, the derived threshold for stable coexistence of supercooled liquid and ice is consistent with both measured and modeled clouds (Korolev, 2007; Desai et al., 2019). The study therefore improves the understanding of the glaciation in turbulent, mixed-phase clouds. Above certain concentrations of ice, the microphysical properties may follow the trend of a purely ice cloud, as predicted by the bulk model and confirmed by the LES results. However, liquid cloud droplets may persist in the locally supersaturated region before they are mixed by turbulence and evaporated. In the atmosphere, liquid droplets in mixed-phase clouds may be maintained by cloud-top (radiative) cooling as well as by a water vapor flux from cloud base, even though the bulk cloud properties may resemble those of an ice cloud. Thus, a lower ice mass fraction than 0.9 may be considered to distinguish ice clouds from mixed-phase clouds. For instance, when a cloud exhibits an ice mass fraction of 0.7, its temperature and supersaturation may be more sensitive to the addition of INP than those in a mixed-phase cloud with an ice mass fraction of 0.5.

In addition, challenges remain in accurately simulating the cloud chamber environments. The side wall wetness can influence the amount of ice number concentration needed to glaciate the cloud, but measuring wall wetness is challenging in the laboratory. Although LES can match the mean droplet radius and number concentration to determine the initial and boundary conditions, the resulting temperature is lower than that measured in the laboratory. One possible reason is the differing roughness lengths on the frozen and liquid water-covered surfaces. Furthermore, bin microphysics scheme suffers from numerical diffusion and leads to broader widths of droplet and ice distributions compared to those observed in the laboratory. As a result, the total particle distribution cannot distinguish the two distinct particle size peaks as observed. The recent decade has seen an increasing application of the Lagrangian particle method (e.g., Shima et al., 2009; Li et al., 2017, 2022; Chen et al., 2021; Yang et al., 2023). The method either tracks super-droplets or individual real particles in a Lagrangian frame and therefore does not have numerical diffusion problems. It is recognized as a better way to physically represent the evolution and transportation of cloud particles, and we expect that may become even more important when collisional growth is considered. In summary, reducing these numerical and instramentation uncertainties to provide more accurate comparisons between models and laboratory experiments is a continuous effort.

*Code and data availability.* The SAM model was kindly provided by Prof. Marat Khairoutdinov of Stony Brook University and publicly available at http://rossby.msrc.sunysb.edu/~marat/SAM.html. The outputs from the bulk model and SAM simulations are stored on NERSC HPSS storage system at https://portal.nersc.gov/archive/home/w/wang1202/www/Wang2024ACP_Glaciation/.

## Appendix A: Sensitivity of LES result to cutoff radius

This study uses a cutoff radius of 3.5 $\mu$m for analyzing droplets and ice crystals to compare the results with those observed by Desai et al. (2019). However, the smallest radius in the bin microphysics is 2 $\mu$m, which may yield more accurate results when compared with the theoretical values. Thus, Fig. A1 presents the results obtained by reducing the cutoff radius to 2 $\mu$m (i.e.,



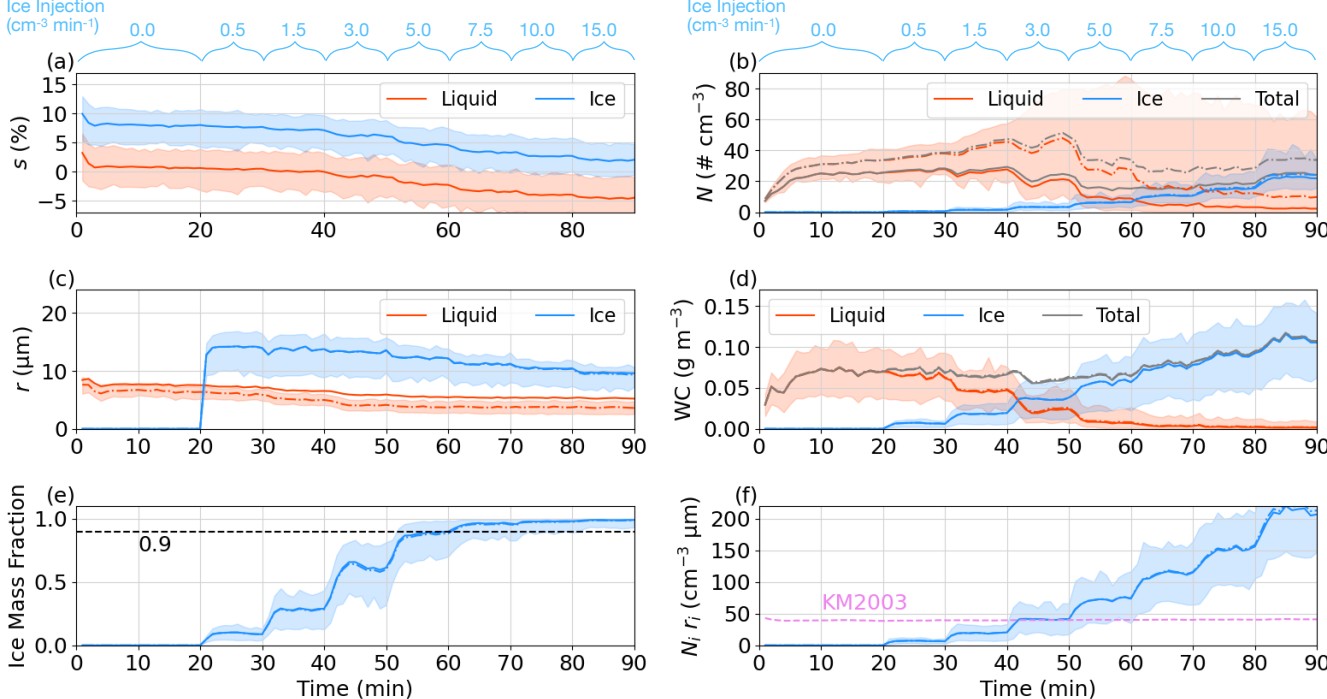

**Figure A1.** Same as Fig. 4c–h, but including the results when the cutoff radius is 2 $\mu$m (dash-dotted line) instead of 3.5 $\mu$m (solid line) for comparison. The shading areas now represent the uncertainty of the dash-dotted line. Note that supersaturation is independent of cutoff radius; panel a (which is Fig. 4c) is included here for reference and to maintain the arrangement of the panels.

using the output of the bin microphysics directly). The most significant change occurs in the droplets' number concentration and radius (cf. the dash-dotted lines to the solid lines in Fig. A1b, c). This is because most droplets are near the smallest end when they are nearly evaporated (Fig. 5c). However, due to the small size of those droplets below 3.5 $\mu$m, their inclusion almost does not affect the liquid water content and the resulting ice mass fraction. Since most of the ice crystals are larger than the cutoff radius (Fig. 5d), the impact on the ice properties is negligible.

**Appendix B: Sensitivity to bottom-top temperature difference and mean temperature revealed by the bulk model**

As the bulk model has been shown to agree with LES in terms of trend changes, this study illustrates how the bulk model can be utilized as a powerful tool to quickly examine the potential influences from the experimental setup. Here, the side walls are saturated. To minimize additional uncertainty, the side-wall temperature is set to the mean of the bottom and top walls, $T_m$. We set $T_m$ to be $-6°$C or $-4°$C, respectively. Additionally, we specify the bottom-top temperature difference, $\Delta T$, to be either
20 K or 24 K.



**Figure B1.** Similar to the bulk model part shown in Fig. 2, but here we test the sensitivity of mixed properties to the bottom-top mean temperature ($T_m$) and the temperature difference ($\Delta T$) instead of the side-wall conditions tested in Fig. 2.

Figure B1 shows that increasing the bottom-top temperature difference results in higher supersaturation for both liquid and ice (Fig. B1d), thus enhancing the condensation and deposition rates (Fig. B1c). This leads to increased liquid and ice water contents (Fig. B1e–f) and a higher temperature (Fig. B1a). Although the consumption of water vapor through condensation and deposition is intensified, the water vapor mixing ratio still increases due to its higher average value at the saturated walls with an

increased temperature difference. These overall effects delay the glaciation. When the bottom-top mean temperature is raised, both the mean temperature and the water vapor mixing ratio increase. This causes the supersaturation for ice to decrease. Interestingly, the supersaturation with respect to liquid is slightly lower initially (compared to the cases when temperature difference is 20 K), but the supersaturation drops with the increase of ice at a lower rate, thus leading to delayed glaciation.



*Author contributions.* A. Wang, S. Krueger, S. Chen, M. Ovchinnikov, W. Cantrell, R. Shaw: conceptualization. A. Wang, S. Krueger,
S. Chen: planning. A. Wang: developing LES code, designing and conducting the model experiments, and visualizing the results. S. Krueger: developing the bulk model code, designing and conducting the model experiments. W. Cantrell: providing the Pi Cloud Chamber experiment details. A. Wang: writing - original draft preparation. S. Krueger, S. Chen, M. Ovchinnikov, W. Cantrell, R. Shaw: writing - review and editing.

*Competing interests.* The contact author has declared that none of the authors has any competing interests.

*Acknowledgements.* We thank Dr. Fan Yang at Brookhaven National Laboratory, Prof. Neel Desai at San Jose State University, and Dr. Xi-angyu Li at PNNL for discussions related to this work. This research is supported by the U.S. Department of Energy Office of Science Atmospheric System Research (ASR) project at Pacific Northwest National Laboratory (PNNL). PNNL is operated for the Department of Energy by Battelle Memorial Institute under Contract DE-AC05-76 RL01830. S. Krueger, W. Cantrell, and R. Shaw were supported by the National Science Foundation (NSF) through Grant AGS 2133229. S. Chen was supported by the NSF National Center for Atmospheric Research, which is a major facility sponsored by the U.S. NSF under Cooperative Agreement No. 1852977. This study used resources of the National Energy Research Scientific Computing Center (NERSC), a U.S. Department of Energy Office of Science User Facility located at Lawrence Berkeley National Laboratory. For transparency and following the position statement of the Committee on Publication Ethics (COPE), we acknowledge the assistance of ChatGPT in generating the first draft of data processing code, checking spelling, correcting grammar, along with some refinements for context.



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
