# Peer review of "Glaciation of Mixed-Phase Clouds: Insights from Bulk Model and Bin-Microphysics Large-Eddy Simulation Informed by Laboratory Experiment"

_EGUsphere, 2024_

## Author Response (AR1)

Reviewer #1

**This is the review of the manuscript entitled "Glaciation of Mixed-Phase Clouds: Insights from Bulk Model and Bin-Microphysics Large-Eddy Simulation Informed by Laboratory Experiment" by Wang et al.**

**This study aims to advance our understanding of mixed-phase clouds using two different modeling approaches to reproduce laboratory experiments conducted in the Pi cloud chamber. Specifically, it is aimed to better understand the theoretically predicted glaciation threshold while the cloud is subsaturated with respect to liquid water. In other words, when is a cloud of mixed-phase type or solely consisting of ice. A bulk scalar mixing model and a large-eddy simulation (LES) with bin microphysics are applied to the Pi chamber studies to examine cloud glaciation. The former model approach allows for complete cloud glaciation while the latter one does not due to the continuous liquid droplet production in the warmer region of the cloud chamber.**

**This manuscript was an enjoyable read. The chosen approach/methods and execution seem to be sound. The topic fits within the journal's science areas. I have mostly minor, clarifying, and technical comments and support publication of this work.**

We appreciate the reviewer's constructive comments that have helped us to improve this paper. We have revised the manuscript in accordance with the reviewer comments, and our point-by-point response is provided below. For clarity, we have used the blue bold font to annotate the reviewers' comments.

**Minor comments:**
1. **Line 105-110: Somehow equations, variables, and text are not consistent. "N" is not defined. There is no subscript "p". This is confusing since one reads first the equations; then to rethink subscripts could be cumbersome.**

We have included the definition of N and rearranged the sentences to make the definition clearer. The rephrased sentence now reads:

"*With the subscript indicating whether the particle (p) is liquid (l) or ice (i), $\rho_p$ is the particle density, $N_p$ is the particle number concentration, $s_p$ is the supersaturation with respect to the phase of the particle, and $\bar{r}_p$ is the particle mean radius.*"

2. **Line 140-142: This is a repetition of line 91.**

Line 91 refers to the setup of the bulk model, while lines 140–142 describe the LES configuration, so we believe it is necessary to mention the temperature setup. The rephrased sentence now reads:

"*Temperature boundary conditions follow the specifications of the bulk model and the experimental setup (Desai et al., 2019), with the bottom wall, top wall, and side walls being kept at 4°C, -16°C, and -12°C, respectively.*"

**3. Line 180: In this equation we have small "n_i". Is this the same as the capital "N_i" above?**

Yes, the reviewer is correct. We have corrected the typo.

**4. Line 182: Please elaborate what you mean with "s_1,0 is the initial supersaturation of liquid without aerosols"? Is it the supersaturation with respect to liquid water? This would be the case with either/or aerosols? What do you mean with "without aerosols".**

It is the supersaturation achieved under the same boundary conditions but without aerosols causing the condensation of droplets. We found that the word "*initial*" was misleading, so we have removed it and rephrased the sentence, which now reads:

"*$s_{1,0}$ is the supersaturation achieved with respect to liquid water under the same boundary conditions but without droplets (i.e., no aerosols) and ice, which requires an additional simulation to determine;*"

**5. Line 210-213: The first sentence "Without the replenishment from droplet evaporation, the water vapor tends towards saturation over ice more rapidly…" is not wrong but likely confusing. The following sentence clarifies the situation a bit. Did you mean "Without the replenishment from droplet evaporation, the water vapor is more rapidly depleted, thus reaching quicker saturation,…"?**

"*Without the replenishment from droplet evaporation, the water vapor is more rapidly depleted*" is correct. "*Thus reaching quicker saturation*" sounds inaccurate because saturation is "*approached*" rather than "*reached*," and the approach is from the supersaturation side rather than the subsaturation side. The sentence is rephrased as follows for a better clarity:

"*Without replenishment from droplet evaporation, the water vapor is more rapidly depleted, approaching saturation with respect to ice and enhancing the decrease in ice crystal size (Fig. 2b, d, e).*"

**6. Line 213-215: I do not readily see in Fig. 2h how the integral radius for glaciation predicted by Korolev and Mazin (2003) matches those predicted by the bulk model. Somehow information about N_i has to be given in this discussion? The lines intersect but what does this mean?**

We appreciate the suggestion. We have added an additional sentence about $N_i$ and the intersection of the lines in Fig. 2h for clarification:

"*Specifically, taking the solid lines in Fig.2 as an example (where the side walls are saturated), the liquid is depleted as $N_i$ reaches 10 cm$^{-3}$ (Fig.2e, f), and the ice integral radius also reaches the value predicted by Korolev and Mazin (2003) at this $N_i$ (see the intersection between the blue and purple lines in Fig. 2h).*"

**7. Line 249: I cannot see a decrease in the droplet number concentration after 20 minutes in Fig. 4d. After 30 mins there seems to be a brief dip in concentration.**

The original sentence indicates that "*the droplet number concentration slightly increases at first*", where we did not indicate a decrease right after 20 min. However, we did not describe the dip after 30 min, so we have added a description for the period between 30 min and 40 min to make the process more complete. The sentences now read:

*"When ice is first injected into the domain at t = 20 min, the droplet number concentration slightly increases at first because of a slowdown in droplet sedimentation due to the reduced droplet size (Fig. 4d, e). During t = 30–40 min, the droplet number concentration remains similar to that during t = 20–30 min. After t = 40 min, the droplet number concentration starts to drop rapidly because the mean supersaturation with respect to liquid falls below 0 (Fig.4c)."*

**8. Line 293-296: There is a lot going in this section on which is difficult to follow. Why does the temperature increase in response to the increased ice concentration? With increasing temperature water mixing ratio should increase since e_l increases? Do you mean a decrease in the supersaturation with respect to liquid water?**

Here, the key is "After a mixed-phase cloud is glaciated," so there are no liquid droplets to evaporate and increase the water vapor mixing ratio, even though the saturation vapor pressure ($e\_l$) increases with temperature. The heat is released during the diffusional growth of ice crystals and absorbed during evaporation of droplets. Both effects combine to determine the temperature trend. Now that the liquid droplets have all been glaciated (evaporated), there is no absorption of sensible heat, and thus the temperature increases. Additionally, there is no further evaporation, so the water mixing ratio only decreases owing to the deposition to ice. We have rephrased the sentences for a better clarity:

*"Specifically, after a mixed-phase cloud is glaciated, there is no evaporative cooling, resulting in a more prominent temperature increase in response to the increased ice concentration. The water vapor mixing ratio decreases due to the deposition of ice. In a fully glaciated state, as the ice concentration increases to a higher level, the increased temperature and decreased water vapor mixing ratio lead to substantially lower supersaturation, resulting in a reduced average size of ice crystals."*

**9. "causing the size of the ice crystals to diminish": If more ice crystal form, supersaturation decreases and resulting ice crystals may be smaller compared to a case with constant supersaturation. However, once ice crystals formed, why should the ice crystal size decrease?**

We realize that the word "diminish" may cause some confusion, as we are comparing steady states rather than discussing a transient state. Additionally, the reduced size refers to the mean size, not the smallest injected ice crystals. I.e., the injected smallest ice crystals will grow, but the mean size

of ice crystals in a steady state reaches a lower mean size compared to that with fewer ice crystals. We have rephrased the sentence for a better clarity:

"*In a fully glaciated state, as the ice concentration increases to a higher level, the increased temperature and decreased water vapor mixing ratio lead to substantially lower supersaturation, resulting in a reduced average size of ice crystals. The trend of reduction in mean ice size is steeper than that before the cloud is glaciated.*"

**Technical corrections:**
**1. Line 105: Missing "respectively"?**

Added.

**2. Line 124: The subscript for Greek letter Xi: It looks like "1" but should be "l" or "i"?**

The sentence is rephrased as "*The temperature weakly affects $\xi_l$ and $\xi_i$.*"

**3. Line 303: Formatting of units.**

Corrected.

**The authors use a bulk model and a LES model with bin microphysics to investigate the glaciation process occurring in the Pi Cloud Chamber. Their experiments are setup according to the laboratory experiments. They found that both the bulk model and the LES model can reproduce the laboratory results, and further found that the bulk model allows complete glaciation while the LES model can always sustain some liquid water. This paper is well written and can be accepted after some minor revisions.**

We appreciate the reviewer's constructive comments that have helped us to improve this paper. We have revised the manuscript in accordance with the reviewer comments, and our point-by-point response is provided below. For clarity, we have used the blue bold font to annotate the reviewers' comments.

**1. Line 1: "radiation" to "radiative"?**

We have replaced "radiation forcing" with simply "radiation" for a better clarity. The sentence now reads:

"*Mixed-phase clouds affect precipitation and radiation differently from liquid and ice clouds, posing greater challenges to their representation in numerical simulations.*"

**2. Lines 35-36: The coexistence of ice and liquid does not ensure the occurrence of the WBF process. Please provide further explanations.**

Thanks for the reminder. We missed the condition that the air should be supersaturated with respect to ice and subsaturated with respect to liquid water. The sentence has been rephrased as follows:

"*When the air is supersaturated with respect to ice and subsaturated with respect to liquid water, as ice and liquid particles coexist, the ice gains mass from the droplets indirectly via the deposition of vapor evaporated from the droplets.*"

**3. Line 78: "near to" to "near"?**

Corrected.

**4. Line 112: Please briefly explain the assumptions behind the "mean equilibrium radii".**

It means that in steady state, with the increase and growth of particles balanced by loss through sedimentation, an equilibrium droplet size distribution is achieved. The distribution has the mode

and mean droplet radii given by the equations. We have rephrased the sentence to briefly explain the assumption as follows:

"*Lastly, for uniform supersaturation and no curvature or solute effects on particle growth, the mean equilibrium radii (the mean particle radii of the distribution achieved as the increase and growth of particles are balanced by loss through sedimentation) for droplets and ice crystals are given by Krueger (2020): ...*"

**5.   Line 182: Please mention that tau_t only consider processes other than microphysics.**

We appreciate the reminder. We have rephrased the sentence as follows:

"$\tau_t$ *is the turbulent mixing time for the flow properties to relax to an equilibrium state (without considering microphysical processes);*".

**6.   Eq. (14): What is phi?**

We have added "*where $\phi$ represents T or $q_v$*".

**7.   Fig. 3: Please add some 2D cross sections along with the 3D isosurfaces. It is not easy to interpret the 3D isosurfaces, at least for this reviewer.**

Thanks for the suggestion. The 2D cross sections are included, and the figure now looks as follows:

[Figure]

**Figure 3.** A snapshot at $t = 50$ min showing the large-scale circulation and its effect on the distribution of flow properties within the chamber. The 3D isosurfaces (a–b) depict warm air at $T = -6°C$ (red-to-yellow color), liquid water content [purple-to-white color in panel (a)], and ice water content isosurface [blue-to-white color in panel (b)]. Gradient colors represent different heights, with lighter colors indicating distance closer to the chamber top, to enhance the 3D visualization. The isosurfaces are opaque, causing some warm air isosurfaces to be obscured by the liquid water and ice crystal isosurfaces. The central slices of panels a–b at y = 1 m are displayed in panels c–d, respectively, with shading indicating temperature and contour lines representing liquid water content (c) or ice water content (d). The solid contour lines indicate above-average values, and dotted contour lines indicate below-average values

**8.  Line 237: Why excluding the regions within 6.25 cm from the wall?**

The purpose is to exclude the regions highly affected by the walls for a better comparison with the bulk scalar mixing model and the atmospheric conditions. We have added an additional paragraph in Section 2.2 (when introducing LES) to clarify this:

*"Additionally, to be able to compare with the bulk scalar mixing model, and to be relevant to the atmospheric conditions, we will mainly focus on the relatively well-mixed region far from the walls (though the domain is not completely mixed). Specifically, the near-wall regions have a sharp gradient in temperature and humidity due to the strong fluxes from the wall, and thus they display*

*different physical characteristics than the center of the chamber. Therefore, we will exclude the near-wall region (within 6.25 cm from the walls) when computing flow properties."*

**9. Line 246: not interquartile.**

We have replaced "interquartile" with "uncertainty".

**10. Fig. 5: Please explain why both (a) and (b) are necessary. What is the dashed line?**

Panel a, showing the probability density, is for comparison with the PDF presented by Desai et al. (2019). Panel b, showing the exact values, illustrates the composition of liquid (shown in panel c) and ice (shown in panel d). The red dashed lines represent the cutoff radius of 3.5 μm. We have included this information in the caption of Fig. 5. The updated caption is attached below:

[Figure]

**Figure 5.** (a) The probability density function of the PSDs, including both ice and droplets (normalized by the particles larger than the cutoff radius of 3.5 $\mu$m), (b) the size distribution of drop and ice, (c) the droplet size distribution, and (d) the ice size distribution. The red dashed lines represent the cutoff radius of 3.5 $\mu$m. Panel a can be compared with the probability density shown in Desai et al. (2019), and panel b reveals the total amount that liquid (c) and ice (d) contribute.

**11. Lines 272-274: It is hard to draw the conclusion from Fig. 6, either because further elaboration is required or because a better-designed figure is required.**

We appreciate the suggestion. The original sentences read:

"*The high ice water content (represented by the solid isoline) arises from the near-bottom region with high supersaturation (representing the updraft regions) and extends to the core region with low supersaturation. This pattern possibly implies how the water vapor in the core region is consumed by the ice.*"

Due to the chaotic nature of the turbulent flow, we agree that the conclusion is not easily shown and may not be robust. Given that "*This pattern possibly implies how the water vapor in the core region is consumed by the ice*" is not the main takeaway and does not influence the main conclusion of this work, we have decided to remove this sentence to avoid confusion. We have also replaced "*arises from*" with "*rises up from*" for better clarity. Additionally, we have improved the clarity of the figure by modifying the increments of the isolines. The revised figure with its caption is attached below:

[Figure]

**Figure 6.** The ice mass concentration (represented by isolines; unit: $g\ m^{-3}$) where solid lines indicate above-average values, and dotted lines indicate below-average values, alongside supersaturation with respect to liquid (indicated by shading), observed at the end of each time period. Red shading represents supersaturation, while blue shading indicates subsaturation. For clarity of the isolines, the increments are 0.01, 0.02, and 0.03 $g\ m^{-3}$ for the first row ($t = 20$–40 min), second row ($t = 50$–70 min), and third row ($t = 80$–90 min), respectively.